# Early-Phase Drive to the Precursor Pool: Chloroviruses Dive into the Deep End of Nucleotide Metabolism

**DOI:** 10.3390/v15040911

**Published:** 2023-03-31

**Authors:** David D. Dunigan, Irina V. Agarkova, Ahmed Esmael, Sophie Alvarez, James L. Van Etten

**Affiliations:** 1Nebraska Center for Virology, Department of Plant Pathology, University of Nebraska-Lincoln, Lincoln, NE 68583-0900, USA; 2Botany and Microbiology Department, Faculty of Science, Benha University, Benha 13511, Egypt; 3Proteomics and Metabolomics Facility, Nebraska Center for Biotechnology, University of Nebraska-Lincoln, Lincoln, NE 68588-0665, USA

**Keywords:** algae, virus, chlorovirus, giant virus, pyrimidine biosynthesis, nucleotide biosynthesis, metabolic reprogramming, genome-to-phenome, augmenting/auxiliary metabolic genes

## Abstract

Viruses face many challenges on their road to successful replication, and they meet those challenges by reprogramming the intracellular environment. Two major issues challenging Paramecium bursaria chlorella virus 1 (PBCV-1, genus *Chlorovirus*, family *Phycodnaviridae*) at the level of DNA replication are (i) the host cell has a DNA G+C content of 66%, while the virus is 40%; and (ii) the initial quantity of DNA in the haploid host cell is approximately 50 fg, yet the virus will make approximately 350 fg of DNA within hours of infection to produce approximately 1000 virions per cell. Thus, the quality and quantity of DNA (and RNA) would seem to restrict replication efficiency, with the looming problem of viral DNA synthesis beginning in only 60–90 min. Our analysis includes (i) genomics and functional annotation to determine gene augmentation and complementation of the nucleotide biosynthesis pathway by the virus, (ii) transcriptional profiling of these genes, and (iii) metabolomics of nucleotide intermediates. The studies indicate that PBCV-1 reprograms the pyrimidine biosynthesis pathway to rebalance the intracellular nucleotide pools both qualitatively and quantitatively, prior to viral DNA amplification, and reflects the genomes of the progeny virus, providing a successful road to virus infection.

## 1. Introduction

The lifestyle of viruses includes the reprogramming of cell-based metabolic pathways to achieve efficient replication, assembly, maturation and escape. Many viruses achieve high levels of biomass production in the short time-periods of the infection cycle, and this seems to match up biosynthetic pathways with an appropriate substrate. However, very little is known about how giant viruses accomplish metabolic adjustments and optimizations.

Among the first known and highly characterized giant viruses are the genus *Chlorovirus* (phylum: *Nucleocytoviricota*; class: *Megaviricetes*; order: *Algavirales*; family: *Phycodnaviridae*) that infect certain eukaryotic green algae having mutualistic lifestyles as endosymbionts of various protists and metazoans [1,2]. These algae are referred to as zoochlorellae [3], and some have been cultured independently of their symbiotic host. This has been essential in evaluating the replication cycle of the chloroviruses, which have an acute lifestyle history strategy with lysis as the phenotype [4,5]. Chloroviruses are ubiquitous in natural waters and can be found in relatively high abundances, occasionally exceeding 10^5^ plaque-forming units (pfu)/mL, but titers fluctuate with the seasons and are more commonly found at 10^1^–10^2^ pfu/mL [6].

Chloroviruses infecting zoochlorellae replicate on the order of hours, with burst sizes up to 1000 particles per cell [7]. The efficiency of the replication strategy would appear to be hampered by the fact that the G+C content of the host cell DNA is significantly higher than that of the virus. In the case of *Chlorovirus* PBCV-1, the viral genome consists of a G+C content of ~40% [8], while the known host (*Chlorella variabilis* NC64A; phylum: *Chlorophyta*) has a nuclear haploid genome consisting of 66% G+C [9]. Assuming cellular replication is somewhat optimized for the G+C content of 66%, we expect that the nucleotide pools are likewise optimal for this cellular characteristic. That is, the cellular G+C nucleotide pool size is greater than the A+T nucleotide pool size in the absence of virus infection, but these values have not been reported.

The replication time for PBCV-1 is 6–8 h post-infection (p.i.), with the early phase (pre-genome replication) occurring up to 60–90 min p.i. [10], during which the virus tends to reprogram the cell for efficient virus production of a burst size of approximately 1000 particles, of which 20–30% form plaques [7]. Electron micrographs indicate that almost all of the particles contain DNA [11]. The PBCV-1 genome is 331 kbp [2]. The haploid nuclear genome of the host is approximately 46 Mbp [9]. Thus, in the 6–8 h period to fully replicate and release PBCV-1 progeny, we estimate that the total amount of DNA in the infected cell goes from 46 × 10^6^ bp at t = 0 min p.i. to ~331 × 10^6^ bp by the end of the infection cycle, consisting of 1000 particles (0.331 × 10^6^ bp/particle). This is an approximately seven-fold increase in DNA, with a bias towards a G+C content of nearly 40%. Measured amounts of adenine incorporation into DNA indicated the total DNA achieved approximately a four-fold increase within 4 h p.i. [12], but these measurements may be skewed towards a low estimate due to altered secondary transport rates, with infection resulting from a rapid loss of membrane potential [13].

Host cellular chromosomes degrade rapidly during PBCV-1 infection, such that by 10 min p.i., no chromosomal DNA is greater than 50 kbp [14]. The extent to which host DNA and/or RNA is recycled during infection is unknown, but even if all the host DNA-derived nucleotides are recycled, there would need to be a multi-fold increase in deoxyribonucleotides to account for the viral synthesis de novo. The same problem exists for RNA synthesis. Where do the nucleotides come from, and what is the G+C to A+T balance needed to accommodate this qualitative and quantitative change in the nucleic acid state of the infected cell?

In this study, we evaluated the PBCV-1 genome for functional annotations of predicted protein-coding sequences (CDSs) that would suggest how and when the virus augments nucleic acid biosynthesis. Then, targeted metabolites (nucleotide biosynthesis pathway intermediates) were evaluated in the context of the early phase of infection, to address the hypothesis that chloroviruses adjust nucleotide pool balances to achieve efficient replication by reprogramming the cellular pyrimidine biosynthetic pathway with virus-encoded proteins.

## 2. Materials and Methods

### 2.1. Genomics and Transcriptional Profiling

*Chlorovirus* PBCV-1 [8] (RefSeq NC_000852.5, Assembly: GCA_000847045.1) and its host *Chlorella variabilis* NC64A [9] (RefSeq Assembly: GCF_000147415.1), and BioProjects PRJNA223657, PRJNA45853 (Taxonomy ID: 554065) genomic features were used to evaluate the nucleotide biosynthesis pathways using the KEGG Mapper tool [15]. The identified genes and associated K numbers are provided in the Appendix A. Manual curation was used to match the viral gene for aspartate transcarbamoylase (ATCase) to the host homolog (E.C. 2.1.3.2). A KEGG pathway map for pyrimidine biosynthesis was annotated with color to indicate virus and host homologs (Figure 1). This map was used to further demonstrate predicted changes in metabolite fluxes.

Transcriptional profiles for all viral genes during the early phase of infection were reported previously using an mRNA-seq approach [16]. Viral mRNA abundances are reported as the median reads per nucleotide (MRPN), and each gene’s expression profile is reported using a K-means clustering method. Each gene’s expression profile was assigned to one of six K-means clusters that are separated into two broad temporal expression profiles, early genes and late genes. Early genes are in K-means clusters 1.1, 1.2 and 1.3 for the transcripts that peak before 60 min p.i.; early-late genes are in K-means clusters 2.1, 2.2, 2.3 for transcripts that do not peak before 60 min p.i. (Table 1).

### 2.2. Targeted LC-MS/MS Metabolomics

#### 2.2.1. Infection Parameters

The algal host *C. variabilis* NC64A was grown in Bold’s basal medium (BBM) modified by the addition of 0.5% (*w*/*v*) sucrose and 0.1% (*w*/*v*) peptone (MBBM) [4]. The production and purification of PBCV-1 virus have been described elsewhere [4,14,17]. Chlorella cells were harvested from 4-day-old cultures (1.5–2.0 × 10^7^ cells/mL) by centrifugation at 4000× *g* for 3 min and then re-suspended in MBBM at a concentration of 2.1–2.5 × 10^8^ cells/mL. PBCV-1 (suspended in the virus stabilization buffer (VSB), 50 mM Tris HCl, 10 mM MgCl_2_, pH 7.8 at a concentration 6.7 × 10^9^ plaque-forming particles (pfu)/mL) was added at a multiplicity of infection (MOI) of 5 pfu/cell.

#### 2.2.2. Sample Collection and Processing

Infected chlorella cells (30 mL) were collected in centrifuge tubes (kept on ice) and centrifuged at 1900× *g* for 3 min; then, the supernatant was removed, cells were re-suspended in 1.5 mL of ice-cold BBM, transferred to 2 mL microcentrifuge tubes and spun for 1 min at 20,000× *g*. The supernatant was removed and spun for an extra 10 s to remove any remaining liquid. The cell pellet weight was recorded, 150 µL of ice-cold methanol was added, and the tubes with samples were placed in liquid nitrogen. Samples were stored at −80 °C. Control (mock-infected cells) had an equal amount (5 mL) of VSB.

#### 2.2.3. Detection and Quantification of the Metabolites

Sample preparation. Metabolites were extracted from the pellets using glass beads (212–300 µm) and 600 µL chilled H_2_O: chloroform: methanol (3:5:12 *v*/*v*) spiked with an internal standard mixture of 100 µM each (^13^C_9_^15^N_3_-dCTP; ^13^C_9_-UTP; ^13^C_10_^15^N_2_-dTTP; ^13^C_10_^15^N_5_-dATP; ^13^C_10_^15^N_5_-dGTP; ^13^C_10_-ATP and ^13^C_10_-GTP). The cells were disrupted with the TissueLyser II instrument (Qiagen) using a microcentrifuge tube adaptor set pre-chilled for 2 min at 20 Hz. The samples were then centrifuged at 16,000× *g* at 4 °C for 10 min, the supernatants were collected, and pellet extraction was repeated once more. The supernatants were pooled, and 300 µL chloroform and 450 µL of chilled water were added to the supernatants. The tubes were vortexed and centrifuged. The upper layer was transferred to a new tube and dried using a speed-vac. The pellets were re-dissolved in 100 μL of 30% methanol. Five µL of the sample was further diluted in 30 µL of 50% acetonitrile.

LC-MS/MS analysis. For LC separation, a Luna-NH2 column (3 µm, 150 × 2 mm, Phenomenex) was used, flowing at 0.88 mL/min. The gradient of the mobile phases A (20 mM ammonium acetate, pH 9.8, 5% acetonitrile) and B (100% acetonitrile) was as follows: 85% B for 1 min, 8% B in 11 min, hold at 8% B for 6 min, then back to 85% B for 0.5 min. The total runtime was 25 min per sample. The LC system was interfaced with a Sciex QTRAP 6500^+^ mass spectrometer equipped with a TurboIonSpray (TIS) electrospray ion source (ESI). Analyst software (version 1.6.3) was used to control sample acquisition and data analysis. The QTRAP 6500^+^ mass spectrometer was tuned and calibrated according to the manufacturer’s recommendations. The instrument was set up to acquire in negative ion mode. The ESI source operation parameters were as follows: source temperature at 450 °C; ion-spray voltage at –4500; ion source gas 1 at 50; ion source gas 2 at 50; curtain gas at 20 psi; collision gas at medium. The metabolites (carbamyl aspartic acid, cytidine, dCDP, dCTP, CDP, CTP, uridine, UMP, UDP, UTP, deoxyuridine, dUMP, dUDP, dUTP, thymidine, dTMP, dTDP, dTTP, dATP, ATP, dGTP, GTP) were detected using Multiple Reaction Monitoring (MRM) transitions that were previously optimized using standards. The MRM transition (Q1-Q3), specific compound settings (declustering potential and collision energy), and retention times for each compound are provided in Appendix A. For quantification, an external standard addition method was used by preparing a series of standard samples containing different concentrations of metabolites into the sample matrix of a pooled sample (to account for matrix effect) and a fixed concentration of the internal standard. The average of the internal standards was used for normalization.

#### 2.2.4. Raw Data

Analytes versus time of infection are available in Appendix A.

## 3. Results

### 3.1. Chlorovirus PBCV-1 Encodes Multiple Pyrimidine Biosynthesis Augmenting Proteins

#### 3.1.1. Chlorella Variabilis NC64A Genome Annotations for Nucleic Acid Biosynthesis Pathways

The alga *C. variabilis* NC64A is the host of the *Chlorovirus* PBCV-1, and its genome was sequenced and annotated [9], providing a foundation for the analysis of the pyrimidine biosynthetic potential of these cells (see Appendix A, for annotations of the pyrimidine metabolism pathway). These data were mapped onto the KEGG Pyrimidine Metabolism Pathway (https://www.genome.jp/kegg-bin/show_pathway?map=map00240&show_description=show, accessed on 11 May 2021). KEGG is the Kyoto Encyclopedia of Genes and Genomes, which was created as a database system for molecular-level information, including biochemical pathways with links to enzyme functions, such as the Pyrimidine Metabolism Pathway [18]. The genomic analysis indicated that many, but not all (46 of 124 enzymatic reactions), of the known associated genes were revealed (https://www.genome.jp/kegg-bin/show_pathway?cvr00240, accessed on 11 May 2021). These 46 algal genes can be seen in Figure 1, highlighted in green. The purine metabolism pathway is similarly covered by the alga (112 of 284 enzymatic reactions) (https://www.genome.jp/dbget-bin/www_bget?pathway+cvr00230, accessed on 11 May 2021).

#### 3.1.2. PBCV-1 Genome Annotations for Nucleic Acid Biosynthesis Pathways

The viral genome annotations indicated that the predicted metabolic reprogramming during virus infection was mostly directed at pyrimidine biosynthesis (Table 1). Only one gene was directed exclusively at purine biosynthesis (phosphoribosyl pyrophosphate synthetase, *pbcv-1_a416r*), while the ribonucleotide reductase complex (RNR) (*pbcv-1_a476r* and *pbcv-1_a629r*) operates in both the purine and pyrimidine biosynthesis pathways. The viral genes are expressed either in the early phase (*pbcv-1_a476r*, *pbcv-1_a629r*, *pbcv-1_a200r*, *pbcv-1_a169r*) or early-late phase of virus replication (*pbcv-1_a551l*, *pbcv-1_a427l*, *pbcv-1_a596r*, *pbcv-1_a674r*, *pbcv-1_a416r*) [16]. Two CDSs are associated with the virion (PBCV-1_A476R and PBCV-1_A629R) [8], and these two proteins make up the functional ribonucleoside-diphosphate reductase (E.C. 1.17.4.1). Four of these genes (*pbcv-1_a169r, pbcv-1_a551l, pbcv-1_a596r, pbcv-1_a674r*) have been confirmed for their predicted functions, with biochemical enzymatic analyses of their recombinant proteins (Table 1); the PBCV-1_A596R protein has a dual function as both a dCMP deaminase and a dCTP deaminase [19]. When superimposed onto the *C. variabilis* pyrimidine metabolism pathway (Figure 1, green boxes), six of the PBCV-1 encoded proteins (green-red boxes) augment the host pathway. In five enzyme reactions, the viral proteins complement the host, i.e., there is no corresponding host protein (Figure 1, red box only). We use a roadway system as a metaphor to describe the elements of this pathway analysis, where we consider metabolites trafficking through a system that has multiple intersections, traffic re-routing options, and traffic-flow modifiers.

### 3.2. By-Passes, Passing Lanes and Traffic Control Predictions

#### 3.2.1. By-Passes: Overriding Regulatory Control and Portals to Other Pathways

The dual-function dCMP–dCTP deaminase (*pbcv-1_a596r*) is a highly conserved gene within the chloroviruses [19]. The protein is not associated with the virion, and the expression pattern indicates it is encoded by an early-late gene, suggesting this function tends to modify cell functions somewhat later in the viral-mediated reprogramming of pyrimidine metabolism (Figure 1F). The activity converts dCMP (E. C. number 3.5.4.12) to dUMP and converts dCTP to dUTP (E. C. number 3.5.4.13) (Figure 1G), both with the release of ammonia. While the cellular pathway provides a gene for the dCMP to dUMP function, there is no cellular gene for the dCTP to dUTP conversion; thus, the virus augments in the first case, and complements in the second case. Additionally, elevated levels of dCTP stimulate the dCMP-to-dUMP conversion caused by the A596R activity, thereby increasing the dTMP pool. However, these dual deamination functions of the A596R enzyme are both inhibited by elevated levels of dTTP, resulting in downstream feedback inhibition of dUTP and dUMP, and, thereby, dTMP accumulation, resulting in the dampening of dTTP trafficking, perhaps associated with the late phase of replication.

Thymidylate synthase X (ThyX) (*pbcv-1_a674r*) (E. C. 2.1.1.148) is a highly conserved gene within the chloroviruses. The protein is not associated with the PBCV-1 virion, and the expression pattern by RNA-seq methods indicates it is an early-late gene [16]. However, Northern blot technology indicates the RNA was maximal at 30 min p.i. and not detectable at 90 min p.i. [22]. Currently we have no explanation for the discrepancy between the two methods. ThyX catalyzes reductive methylation of the dUMP to dTMP, as do all thymidylate synthases (Figure 1H); however, the viral enzyme uses FAD as a source of reducing equivalents that are derived from NAD(P)H [22]. While the *C. variabilis* genome contains the ThyA enzyme (E. C. 2.1.1.45), the virus seems to by-pass the cellular contribution to dTMP synthesis when entering late-phase replication with the ThyX activity, resulting in a strong pull to the dTMP precursor that traffics to the dTTP pool through gene augmentation.

#### 3.2.2. Passing Lane: Viral Enzymes That Augment Cellular Function

The genes coding for ribonucleoside diphosphate reductase (RNR) [E.C. number 1.17.4.1] are highly conserved in many dsDNA viruses and are 100% conserved in the chloroviruses. The PBCV-1 RNR is a virion-associated protein, and the two genes are coordinately expressed early. The enzyme converts ribonucleoside diphosphate to deoxyribonucleotides (Figure 1A). With respect to the pyrimidine biosynthesis pathway, the RNR acts as a passing lane, where CDP is converted to dCDP, and UDP to dUDP, augmenting a cellular homolog. The purine nucleoside diphosphates are also converted to their corresponding deoxy counterparts with this function. Additionally, the enzyme is not predicted to convert triphosphate substrates UTP and CTP to dUTP and dCTP (E. C. 1.17.4.2), respectively, where there is no corresponding cellular homolog. The viral enzyme activity is predicted to be an immediate–early event based on the virion association, and the mRNA expression profile is an early gene, as determined with RNA-seq methods [16]. A BLAST search with both the small subunit PBCV-1_A476R protein (RefSeq NP_048832) and the large subunit protein PBCV-1_A629R (RefSeq NP_048985) indicates the substrate is most likely the ribonucleoside-diphosphate, possibly containing a diiron (III)-tyrosyl radical (class Ia) utilizing thioredoxin, which is also encoded by the PBCV-1 genome (PBCV-1_A427L). The RNR activity tends to re-direct nucleotide pools to a DNA precursor, and this is likely a consequence of the salvage pathway from RNA degradation that occurs soon after virus infection [23].

Deoxyribonucleoside kinase (*pbcv-1_a416r*) (E. C. 2.7.4.14) is conserved in the chloroviruses but not entirely; notably, most viruses in the subclade NC64A-I [24] do not carry this homolog, whereas viruses in subclade NC64A-II, Pbi and SAG viruses all have the gene [25]. The protein is not associated with the PBCV-1 virion [8], and it is expressed early-late as assessed with RNA-seq methods [16]. Deoxynucleoside kinases consist of cytidine (EC:2.7.1.74), guanosine (EC:2.7.1.113), adenosine (EC:2.7.1.76) and thymidine kinase (EC:2.7.1.21) (the later enzyme phosphorylates deoxyuridine and deoxycytosine, as well). Production of deoxynucleotide 5′-monophosphate catalyzed by these enzymes from a deoxynucleoside uses ATP and yields ADP in the process. Both CMP and UMP are produced by phosphorylation with similar efficiency by the bifunctional eukaryotic enzyme, where dCMP can be an acceptor. However, the chlorovirus enzyme is most similar to the monofunctional prokaryotic enzymes CMP kinase (E.C. number 2.7.4.25) and UMP kinase (E.C. number 2.7.4.22). The nucleoside/nucleotide kinase (NK) protein superfamily consists of multiple families of enzymes with shared structures. Functionally, they are related to the catalysis of the reversible phosphate group transfer from nucleoside triphosphates to nucleosides/nucleotides, nucleoside monophosphates, or sugars. Members of this family play a wide variety of essential roles in nucleotide metabolism, the biosynthesis of coenzymes and aromatic compounds, and the metabolism of sugar and sulfate [8,16,25].

dUTP pyrophosphatase (*pbcv-1_a551l*) (E. C. number 3.6.1.23) (Figure 1D) is a highly conserved gene in the chloroviruses. The protein is not associated with the PBCV-1 virion [8] and is expressed maximally at 30 min p.i., as demonstrated with Northern blot methods [21], and is an early-late gene [16]. The A551L protein is prominent throughout the late phase of infection, as measured with Western blotting methods [21]. The recombinantly cloned and expressed protein has the predicted biochemical function, dUTP pyrophosphatase [21]. This event predicts an enhanced overall flux from CTP to dCTP to dUTP to dUMP, such that dUMP is a node for flux to dTTP.

dCMP deaminase (*pbcv-1_a596r*) (E. C. 3.5.4.12) is a highly conserved gene in the chloroviruses. The protein is not associated with the PBCV-1 virion and is expressed early-late in the infection cycle, as determined with RNA-seq methods [16], and the mRNA abundance is maximal at 90 min p.i., as demonstrated with Northern blotting methods [19], indicating that this function is more abundant as the replication approaches viral DNA synthesis initiation. The enzyme is bifunctional (as described below), but as a traffic by-pass step, it deaminates dCMP to form dUMP with the release of ammonia (Figure 1F) [19,26].

#### 3.2.3. Traffic Control: Feedback Control Loops to Speed Up or Slow Down Metabolic Fluxes

A gateway for pyrimidine biosynthesis de novo is through the complex and sensitive multi-enzyme complex referred to as CAD: carbamoyl-phosphate synthetase 2/aspartate transcarbamylase/dihyroorotase. This enzyme complex conducts the first three steps toward pyrimidine biosynthesis de novo and is sensitive to nucleotide pool concentrations, both positively and negatively (Figure 1C). The cellular enzyme is composed of catalytic and regulatory subunits [27]. The enzyme is allosterically regulated by the binding of CTP to the regulatory subunits, causing inhibition of enzyme velocity. The binding of UTP enhances the binding of CTP and results in further inhibition of the enzyme, whereas the binding of UTP alone has little effect. The converse is true when ATP is bound to the regulatory subunit, causing the enzyme to be maximally active. Thus, high flux rates of the pyrimidine biosynthesis pathway toward the production of CTP and UTP are attenuated by feedback regulation, eventually [28]. However, *Chlorovirus* PBCV-1 only encodes the catalytic subunit of ATCase (PBCV-1_A169R) and presumably is insensitive to the feedback inhibition affected by CTP and UTP [20]. Though not associated with the PBCV-1 virion, the early expression of the viral ATCase catalytic mRNA is maximal by 30 min p.i. and declines thereafter, as determined by Northern blotting [20] and confirmed by RNA-seq methods [16]. The early expression of an ATCase catalytic subunit and no corresponding regulatory subunit predicts that pyrimidine biosynthesis shifts to an “all-on” mode early in the infection cycle, with little or no feedback inhibition to slow the flux rate, and by-passes the host enzyme, as suggested by the blue arrow in Figure 2. This was demonstrated biochemically where there was a two-fold increase in ATCase activity in virus-infected cell-free extracts starting at 30 min p.i. and lasting until 3 h p.i., compared to mock-infected cell-free extracts, as measured by the conversion of L-[U-^14^C]aspartate to carbamyl [U-^14^C]aspartate [20]. The viral ATCase gene (*pbcv-1_a169r*) is not conserved in the chloroviruses but is found in those members of the *C. variabilis* NC64A-II subclade viruses only. This protein matches best to one of the two *C. variabilis* NC64A homologs (NCBI-Protein ID: XP_005848545.1; identities = 60%, Bit score = 308).

The dCMP/dCTP deaminase (*pbcv-1_a596r*) is a highly conserved gene within the chloroviruses, as described above. The chlorovirus dCMP/dCTP deaminases are sensitive to nucleotide pool changes. The viral dCTP deaminase complements the cell (E. C. 3.5.4.13), while the viral dCMP deaminase augments the cell (E. C. 3.5.4.12), and both of the viral activities are negatively regulated by increasing levels of dTTP (Figure 2, red dashed line with a minus sign “−”). The dCMP deaminase is positively regulated by increasing levels of dCTP (Figure 1, red solid line with a plus sign “+”). However, these influences tend to push dCTP toward an increased dTMP pool that feeds to dTTP, where dTTP tends to down-regulate the dCTP to dUTP and CMP to UMP conversions. That is, dTTP taps the brakes on the viral dCTP/CMP deamination activity.

The virus-encoded phosphoribosylpyrophate synthetase (E.C. 2.7.6.1; PBCV1_A438L) of the purine biosynthesis pathway produces the same product from the pyrimidine biosynthesis pathway with uridine monophosphate synthetase: 5-phospho-alpha-D-ribose 1-diphosphate produced from the transfer of pyrophosphate from ATP to ribose 5-phosphate (PRPP). PRPP is one of the four key regulatory modulators of the purine biosynthesis pathway and is an essential component of the purine salvage pathway, as well as the synthesis of purines de novo. Phosphoribosylpyrophate synthetase is not the committed step in purine biosynthesis but is negatively affected by GDP, and at a distinct allosteric site is inhibited by ADP. The enzyme is activated by phosphate. Thus, the virus-encoded enzyme appears to be a portal between the pyrimidine and purine biosynthetic pathways, producing PRPP.

Genomic annotations predict pyrimidine fluxes to be directed toward enhanced dTTP synthesis and less toward dCTP synthesis (Figure 1). This supports the initial hypothesis. The hypothesis was then further evaluated with biochemical assessments of targeted nucleotide biosynthesis metabolic intermediates.

### 3.3. Metabolomic Analyses of Targeted Pyrimidine Biosynthetic Pathway Intermediates

From genomic and transcriptomic studies, we hypothesized that the nucleotide pools adjust to a state that favors the synthesis of viral nucleic acids, which is approximately 40% G+C, by the onset of viral DNA synthesis. A targeted liquid chromatography—tandem mass spectrometry (LC-MS/MS) metabolomic approach was developed to focus on the pyrimidine biosynthesis pathway intermediates during the early phase of infection (0–90 min p.i.). All flux patterns of the measured metabolites were complex, with either second- or third-order kinetics at the time of evaluation (Appendix A, Metabolite flux patterns by kinetic order). Additionally, a few purine nucleotides were evaluated as well (ATP, dATP, dGTP). This analysis indicates that pyrimidine biosynthetic pathway intermediates change little during the immediate–early phase of infection (0–20 min p.i.) but are altered significantly at the time of DNA synthesis initiation (40–60 min p.i.), as indicated by multi-fold changes (Table 2).

Additionally, the analysis indicates that the pyrimidine biosynthesis pathway is up-regulated, generally, where most metabolites increase several-fold during the early phase up to 60 min p.i. (for the comprehensive set of kinetic analyses for all measured metabolites see Appendix A for the measured value of each replicate). Approximately half of the measured metabolites tend to decrease in abundance after 60 min p.i. (third-order kinetics).

However, in the immediate–early phase (0 to 20 min p.i.) most metabolites decline in abundance, then rebound (Appendix A). Of particular interest, the dCTP:dTTP at t = 0 min p.i. (dCTP:dTTP_t = 0_ = 39 zmol/cell: 9.5 zmol/cell) was approximately 4.5. This ratio is biased, seemingly to support a high G+C content of the organism (66%). As the infection progresses, there is a slow but steady increase in most of the pyrimidine intermediates, including both dCTP and dTTP, but at varying rates (Figure 3). By 40 min p.i., the dCTP:dTTP was approximately 0.6, resulting in a more than seven-fold change, biased to the lower state of G+C content of the virus.

The measured fluxes appear to derive from a high abundance of uridine and UMP, each at >10^6^ zmol/cell (Appendix A). Surprisingly, two intermediates were measured but not detected throughout the infection: N-carbamoyl-L-aspartate and dUTP. This is consistent with a high flux rate of this substrate but apparently not influenced by infection. The flux rate for thymidine and derivative nucleotides increased by 14 min p.i., before other intermediates started to increase (Table 2)

At 40 min p.i., uridyl-nucleotides remain suppressed, while cytidine and thymidine nucleotides are significantly above the initial state. At 60 min p.i., all pyrimidine nucleotides were above the initial state.

The purine nucleotides evaluated in this study (ATP, dATP, dGTP) were somewhat less affected by the infection progress (Appendix A, and Table 2). ATP and dGTP had relatively high initial states that decreased with infection and did not recover to the initial state, even by 90 min p.i. (>800 zmol/cell). dATP had a relatively low initial state (12 zmol/cell) that remained relatively unchanged during the first 20 min p.i., but then increased somewhat by 40 min p.i. (~80 zmol/cell).

The dCTP:dTTP at t = 60 min p.i. (dCTP:dTTP_t = 60_ = 125 zmol/cell:185 zmol/cell) was 0.67, where the abundances were maximal. This ratio is biased to support a high A+T content of viral DNA that is just beginning to be synthesized at this point in the infection and supports the initial hypothesis. In general, the pyrimidine intermediates were maximal at 40–60 min p.i., after which the abundances declined somewhat (Appendix A). The notable exceptions to this trend were the uridylate nucleotide intermediates, which tended to continue to increase in abundance beyond 90 min p.i. (Appendix A).

## 4. Discussion

Viruses are obligate intracellular symbionts, and they tend to pirate cellular functions for their own means. They manipulate host cell metabolism, including glycolytic, amino acid, fatty acid and nucleotide metabolism [29,30]. For example, both enterovirus and human cytomegalovirus infections were shown to markedly increase the metabolites in glycolysis, TCA cycle and pyrimidine biosynthesis pathways with corresponding up-regulation of specific transcripts [31,32]. Not surprisingly, these pathways become useful targets for antiviral therapies [31,33].

Nucleotide pools are potentially a rate-limiting factor in virus replication, especially for acute infections with large burst sizes. If the nucleotide composition ratio (i.e., G+C %) of the virus matched that of the host, there would be a need to increase nucleotide biosynthetic rates but not shift away from the host ratios. This type of increase in nucleotide pools might be achieved with the augmentation of rate-limiting enzymes by viral homolog addition. However, if the nucleotide pool demand by the virus significantly differs from the host ratios, pathway fluxes would need to be (i) altered by post-translational modification of the host enzymes, or (ii) complementation with viral enzymes, or (iii) or both.

Chloroviruses are large dsDNA viruses that replicate on the order of hours, with burst sizes up to 10^3^ particles per cell, resulting in a multi-fold increase in the total DNA of the cell. The hosts tend to have significantly higher G+C content in their nucleic acids than do the chloroviruses. We addressed the question of how does the infected cell adjust to an increased demand for nucleotide precursors to fulfill the need for viral RNA and DNA synthesis, as well as the crossover to a different balance of G+C content by the dominate virus replication machinery.

We hypothesized that chlorovirus-encoded protein functions alter the nucleotide biosynthesis pathway to reset the nucleotide balance while boosting the overall pool sizes, and that this alteration is pointed at the pyrimidine biosynthesis pathway primarily. This hypothesis was supported by the evaluation of the (i) genomic data of the type member of the genus *Chlorovirus*, PBCV-1, (ii) transcriptional profile data of a PBCV-1 infection, and (iii) analyte fluxes of the nucleotide biosynthesis pathways during the early phase of virus infection using a targeted metabolomics analysis. PBCV-1 genes either augmented or complemented the pyrimidine biosynthesis pathway, primarily, but also appeared to alter synthesis by the addition or subtraction of regulatory functions, such as the virus-encoded ATCase that lacks regulatory subunits (Figure 2). The onset of virus gene transcriptional profiles coincided with NTP and dNTP precursor accumulations, such that by 60 min p.i., dNTP pools were maximum, in time for the onset of viral DNA synthesis. Host transcriptional processes during this early phase were estimated by evaluating the mRNA half-lives of the pyrimidine biosynthesis genes (Appendix A). Most, but not all, of the host mRNAs had reduced abundances relative to the time = 0 control, with half-lives in the 25–50 min range. Two notable exceptions were the nucleoside-diphosphate kinase (E.C. 2.7.4.6; t_1/2_ = 581 min) and 5′nucleotidase (E.C. 3.1.3.5; t_1/2_ = 186 min), suggesting some need for these enzymatic functions in the late phase of infection. This degradation of host pyrimidine biosynthesis mRNAs was accelerated relative to the majority of host mRNAs (mean t_1/2_ approximately 80 min). Notable exceptions with relatively long half-life values were mRNAs for protein synthesis (e.g., ribosomal proteins and translation elongation factors), presumably to support virus replication [34]. How degradation of different cellular mRNAs is controlled is unknown.

Most of the genes involved in the nucleotide biosynthesis process are conserved across a diverse set of chloroviruses, with notable exceptions [25]. Within the conserved set are members of gene gangs. Chlorovirus gene gangs are conserved clusters of collinear monocistronic genes, and the chloroviruses have 25 gangs consisting of three or more members [24]. The evolutionary selection of the gene gangs infers a common metabolic or structural function required by the virus, not unlike a bacterial operon. In some cases, the gang function is readily inferred by the annotations of the members; in other cases it is much less clear. Nevertheless, the presence of a snitch (member with known function) infers a guilt-by-association assignment of presumed function to the gang. In each case, the nucleotide biosynthesis genes of the chloroviruses with known functions suggest that these gene gangs are contributing to nucleotide pool alternations observed in this study, although the connections are not always apparent. Snitches were apparent in gene gangs 3, 7, 11, 14, 23 and 25. The virus-encoded large and small subunit RNRs are members of gene gangs 25 and 3, respectively. These are the only polypeptides involved in nucleotide biosynthesis that are associated with the virion [8], suggesting the ribonucleotide reduction to the deoxy forms is selected to be active immediately upon virus entry, perhaps contributing to the salvage pathway [23].

In several respects, chlorovirus biology reflects aspects of certain bacteriophage infections. One case of interest is *Pseudomonas aeruginosa* and the associated myovirus phage PAK_P3, with burst sizes at 10^3^ pfu/cell on the order of 10′s of min p.i. [35]. This infection results in a rapid and drastic decay of host mRNA, replaced by viral transcripts leading to significant changes in pyrimidine metabolism, as well as amino acid, lipidpolysaccharide and nucleotide sugar metabolisms. Interestingly, few host genes are differentially expressed, pointing to the significance of virus-mediated virocell reprogramming through temporally regulated viral gene expression, essentially like the chlorovirus infection dynamic.

In summary, the in silico analysis and the experimental results support the hypothesis that *Chlorovirus* PBCV-1 infection reprograms the pyrimidine biosynthesis pathway during the early phase of infection to reflect the change in the host genome 66% G+C content to the 40% G+C content of the virus genome. We suspect this approach and these findings are generally useful for understanding how obligate intracellular symbionts control the environment of the host cell for their own benefit. This approach also allows for hypothesis development in terms of drug targeting of critical points in the replication pathways of many parasitic agents of interest that depend on intracellular metabolism.

## Figures and Tables

**Figure 1 viruses-15-00911-f001:**
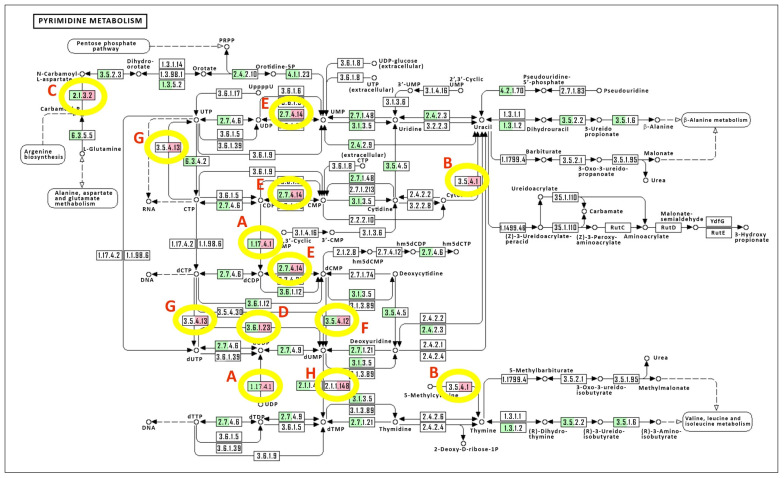
KEGG pathway of the pyrimidine metabolism, annotated with *C. variabilis* NC64A and PBCV-1 genes. From genomic analyses of *C. variabilis* NC64A and the *Chlorovirus* PBCV-1, functional annotations of genes were mapped onto a KEGG (Kyoto Encyclopedia of Genes and Genomes) flow chart for pyrimidine metabolism. The chart indicates the (i) name of a metabolite, with the position indicated by “o”, (ii) enzyme function responsible for a metabolite conversion (annotated by the Enzyme Commission (E.C.) number), (iii) flux direction, indicated by a connecting arrow, and (iv) associated metabolic pathways that feed into or out of the pyrimidine metabolism pathway. Where the *C. variabilis* gene was identified, the E.C. number was color-coded green. Where the PBCV-1 homologous gene was identified, the E. C. number was color-coded red. If both the *C. variabilis* NC64A and PBCV-1 homologous gene was identified, the E. C. number was color-coded green and red. If no gene was identified, the E. C. number was left uncoded and appears as a white box. The information for the genomic analyses of the virus and host cell are found in Appendix A, and the lists of genes used for the KEGG Mapper to the Pyrimidine Metabolism Pathway are found in Appendix A. (**A**) Ribonucleoside-diphosphate reductase (E.C.1.17.4.1): augmenting enzyme of both purine and pyrimidine metabolism pathways. (**B**) Cytosine deaminase (E.C.3.5.4.1): complementing enzyme of pyrimidine metabolism. (**C**) Aspartate/ornithine carbamoyltransferase (E.C.2.1.3.2): complementing enzyme of the pyrimidine metabolism that is not sensitive to nucleotide feedback regulations. (**D**) dUTP pyrophosphatase (E.C.3.6.1.23): augmenting enzyme of the pyrimidine metabolism. (**E**) Deoxynucleoside kinase (E.C.2.7.4.14): augmenting enzyme of pyrimidine metabolism but not purine metabolism. (**F**) dCMP deaminase (E.C.3.5.4.12): augmenting enzyme of pyrimidine metabolism that is sensitive to dCTP positive feedback and dTTP negative feedback. (**G**) dCTP deaminase (E.C.3.5.4.13): complementing enzyme of pyrimidine metabolism that is sensitive to dTTP as a negative feedback. (**H**) Thymidylate synthase X (E.C.2.1.1.148): complementing enzyme of pyrimidine metabolism that may override the host-encoded ThyA; ThyX is FADH2-dependent.

**Figure 2 viruses-15-00911-f002:**
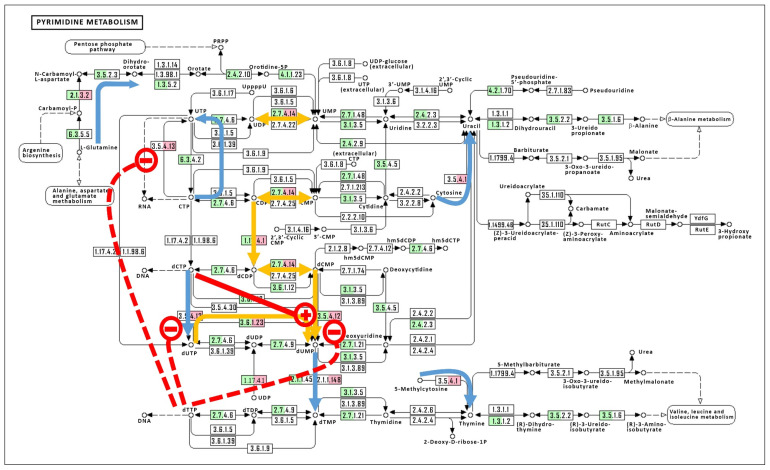
KEGG pathway of the pyrimidine metabolism annotated with *C. variabilis* NC64A and PBCV-1 genes and comprehensive “roadway” predictions. *C. variabilis* NC64A genes are colored green, and PBCV-1 genes are red. This figure summarizes the virus-mediated metabolic fluxes where viral genes either (i) complement the pathway by providing a function not identified in the host genome, and we refer to this as a “roadway by-pass” (indicated by a blue arrow); or (ii) augment the pathway by providing a function that is present in the host genome, and we refer to this as a “roadway passing lane” (indicated by a yellow arrow). Metabolite-directed flux rates of virus-encoded enzymes are sensitive to either positive feedback regulators (indicated by a red solid line with a plus sign “+”) or negative feedback regulators (indicated by a red dashed line with a minus sign “−”).

**Figure 3 viruses-15-00911-f003:**
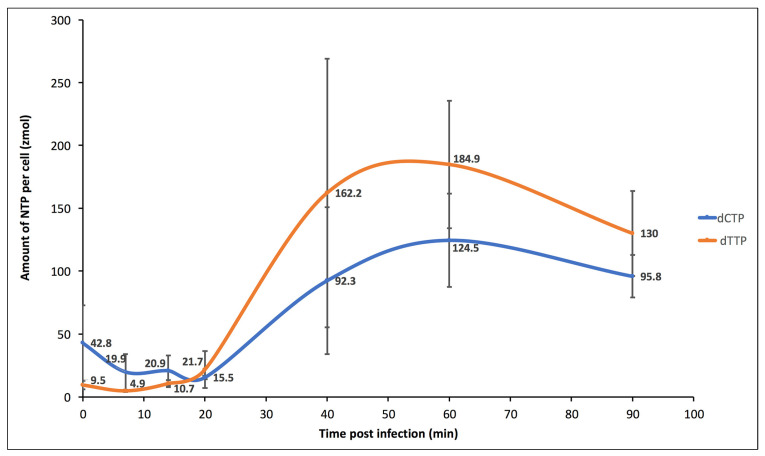
Chart of dCTP and dTTP amounts per cell post infection time. Values represent the mean of three biological replicates. The standard deviation values were at, 0 min post infection time (p.i), 30 and 3.5; at 7 min p.i., 14 and 1; at 14 min p.i., 12.1 and 2.8; at 20 min p.i., 1.5 and 14.7; at 40 min p.i., 58.4 and 107; at 60 min p.i., 37.2 and 50.8; at 90 min p.i., 16.9 and 33.8, for dCTP and dTTP, respectively. A secondary early phase (t = 20 to 60 min p.i.) indicates that uridine and UMP change little, but the extent of change was highly significant in terms of resourcing the pyrimidine biosynthesis pathway. These two metabolites were three orders of magnitude greater than the next most abundant metabolites that we measured in the pyrimidine biosynthesis pathway. Thus, small percentage changes in these two compounds will result in large percentage changes downstream. Uridine increased with time, indicating an influx, likely via uracil from the alanine/aspartate metabolism pathway and the beta-alanine metabolism pathway. UMP decreased early but began to recover by 60–90 min p.i. This suggests there is a delay in flux from uridine, while there is a pull via UDP deoxynucleoside kinase (E.C. 2.7.4.14; PBCV1_A416R) (Figure 1E).

**Table 1 viruses-15-00911-t001:** Functional annotations of PBCV-1 nucleotide metabolism genes, ordered by infection cycle progression.

Functional Annotation	Genes	RefSeq	KEGG Number	EC Number	Reaction (IUBMB) [KEGG Reaction]	Expression Pattern ^a^	Virion Association ^b^
Ribonucleotide reductase (small subunit)	*a476r*	NP_048832.1	K00524	1.17.4.1	2′-deoxyribonucleoside 5′-diphosphate + thioredoxin disulfide + H_2_O = ribonucleoside 5′-diphosphate + thioredoxin [RN:R04294]	early	yes
Ribonucleotide reductase (large subunit)	*a629r*	NP_048985.1	K00524	1.17.4.1	2′-deoxyribonucleoside 5′-diphosphate + thioredoxin disulfide + H_2_O = ribonucleoside 5′-diphosphate + thioredoxin [RN:R04294]	early	yes
Cytosine deaminase	*a200r*	NP_048547.1	K01485	3.5.4.1	cytosine + H_2_O = uracil + NH_3_ [RN:R00974]	early	ND ^c^
* Aspartate/ornithine carbamoyltransferase [20]	*a169r*	NP_048517.1	K11540	2.1.3.2	carbamoyl phosphate + L-aspartate = phosphate + N-carbamoyl-L-aspartate [RN:R01397]	early	ND
* dUTP pyrophosphatase [21]	*a551l*	NP_048907.1	K01520	3.6.1.23	dUTP + H_2_O = dUMP + diphosphate [RN:R02100]	early-late	ND
Deoxynucleoside kinase/Dephospho-coenzyme A kinase/Deoxyadenosine/deoxycytidine kinase	*a416r*	NP_048773.1	K13800	2.7.4.14	(1) ATP + (d)CMP = ADP + (d)CDP [RN:R00512 R01665]; (2) ATP + UMP = ADP + UDP [RN:R00158]	early-late	ND
Thioredoxin	*a427l*	NP_048784.1	K00384	1.8.1.9	2′-deoxyribonucleoside 5′-diphosphate + thioredoxin disulfide + H_2_O = ribonucleoside 5′-diphosphate + thioredoxin [RN:R04294]	early-late	ND
Phosphoribosyl pyrophosphate synthetase	*a568l*	NP_048924.1	K00948	2.7.6.1	ATP + D-Ribose 5-phosphate <=> AMP + 5-Phospho-alpha-D-ribose 1-diphosphate	early-late	ND
* dCMP deaminase [19]	*a596r*	NP_048952.1	K01493	3.5.4.12	dCMP + H_2_O = dUMP + NH_3_ [RN:R01663]	early-late	ND
* dCTP deaminase [19]	*a596r*	NP_048952.1	K01494	3.5.4.13	dCTP + H_2_O = dUTP + NH_3_ [RN:R02325]	early-late	ND
* Thymidylate synthase X [22]	*a674r*	NP_049030.1	K03465	2.1.1.148	5,10-methylenetetrahydrofolate + dUMP + NADPH + H^+^ = dTMP + tetrahydrofolate + NADP^+^ [RN:R06613]	early-late	ND

* Recombinant proteins have the expected activity [Reference]. ^a^ PBCV-1 transcriptional profile [16]. ^b^ PBCV-1 protein profile [8]. ^c^ ND—not detected. Cytosine deaminase (PBCV-1_A200R; E. C. number 3.5.4.1) is not associated with the PBCV-1 virions but is encoded by an early gene. This indicates that the mRNA is synthesized within the first 7 min p.i. and is at a maximal abundance by 20 min p.i. but is greatly diminished by 60 min p.i. The enzyme converts cytosine (Figure 1B, upper circle) or 5-methylcytosine (Figure 1B, lower circle) to uracil or thymine, respectively. There are no known cellular homologs of this gene in the *C. variabilis* genome; thus, PBCV-1 cytosine deaminase is a complement to the host pyrimidine biosynthesis pathway. The viral enzyme acts as a by-pass in the pyrimidine pathway, possibly routing CMP to uridine via the deamination of cytosine and then to uracil. The source of the cytosine is not clear but may be derived from degradation of cellular DNA and/or RNA.

**Table 2 viruses-15-00911-t002:** Heat map of nucleotide metabolite amounts per cell versus time of infection.

	Initial Value (Zeptomole */Cell)	Fold Change Compared to t = 0 mpi(zmol per Cell @ t = x mpi/zmol per Cell @ t = 0 mpi)
Analytes	0 mpi	0 mpi	7 mpi	14 mpi	20 mpi	40 mpi	60 mpi	90 mpi
Uridine	1,010,260.11	1.0	2.2	2.2	2.7	2.5	2.9	3.5
UMP	3,462,328.07	1.0	0.6	0.6	0.6	0.7	0.8	0.9
UDP	248.24	1.0	0.7	0.7	0.6	1.1	1.4	1.9
UTP	56.39	1.0	0.6	0.6	0.5	1.0	1.3	2.0
Cytidine	236.46	1.0	1.5	2.0	2.0	3.4	4.4	4.3
CDP	99.48	1.0	0.6	0.6	0.6	1.5	1.7	1.6
CTP	39.31	1.0	0.6	0.6	0.5	1.3	1.5	1.5
dCDP	7.16	1.0	0.5	0.7	1.4	11.3	15.7	11.3
dCTP	42.75	1.0	0.5	0.5	0.4	2.2	2.9	2.2
Deoxyuridine	14.96	1.0	1.1	3.1	9.8	43.0	40.8	23.1
dUMP	4.50	1.0	0.5	0.9	1.9	9.2	14.9	23.8
dUTP	Bdl ^#^	bdl	bdl	bdl	bdl	bdl	bdl	bdl
Thymidine	12.83	1.0	1.6	6.6	24.2	76.8	57.4	30.0
dTMP	13.54	1.0	0.8	3.0	8.0	18.3	20.5	15.6
dTDP	18.67	1.0	0.6	1.7	4.8	25.4	27.5	17.7
dTTP	9.47	1.0	0.5	1.1	2.3	17.1	19.5	13.7
dGTP	891.04	1.0	0.6	0.6	0.5	0.7	0.7	0.8
ATP	846.96	1.0	0.6	0.6	0.5	0.7	0.7	0.8
dATP	12.20	1.0	0.5	0.7	0.9	6.6	6.8	4.9
N-Carbamoyl-L-aspartate	bdl	bdl	bdl	bdl	bdl	bdl	bdl	bdl

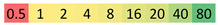
. The color of the cells is changing from red (low value) to green (high value).* zeptomole = 1E-21 mole. ^#^ bdl- below detectable limits.

## Data Availability

All processed data used to develop the presented figures and tables are available in the Appendix A.

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
