# Peer review of "Early-Phase Drive to the Precursor Pool: Chloroviruses Dive into the Deep End of Nucleotide Metabolism"

_viruses, 2023, doi:10.3390/v15040911_

Round 1

Reviewer 1 Report

Dunigan et. al. provides an interesting but incomplete manuscript describing host and viral metabolic pathways to synthesize nucleotides. They also discuss the possible importance of these pathways as rate limiting for viral infections. Unfortunately, while the extensive research into the literature and analysis of genome and transcriptome are highly appreciated, such kind of information should rather be associated with a review and not a primary scientific article. No new data is provided in the first 2/3 of the manuscript but rather a re-analysis of previously published one. I do not consider this manuscript appropriate for a primary scientific article publication. Few extra considerations:

In the introduction it is stated: ¨The efficiency of the replication strategy would appear to be hampered by the fact that the G+C content of the host cell DNA is significantly higher than that of the virus.¨ If there is any evidence that this is true, provide a citation. Otherwise move this apparent conclusion of the author (with no evidence to support it) into the discussion.

Moreover, there is no evidence demonstrating that DNA synthesis is a limiting step in viral replication. If the authors believe this is true, I suggest they perform experiments with sublethal doses of hydroxyurea (or similar) in order to hamper the pool of deoxynucleotides. Inhibitors of pyrimidine and purine nucleotide biosynthesis might also be tested (see 10.3390/microorganisms10081631)

I recommend the authors to clone and express recombinantly some of the enzymes discussed in the manuscript. The enzymatic characterization of viral enzymes is of high interest for biotechnological applications and would increase the impact of the manuscript.

Figure 1, 2 and 3 make the manuscript look rather like a powerpoint presentation and not a scientific article. It is not needed to include 3 Figures with similar information. Please compact it into 1 Figure.

Recycling of nucleotides incorporated into host DNA can be studied by isotope labelling. I suggest the authors explore such possibilities.

The authors speculate about fluxes multiple times using the data provided in Table 2. The data provided in Table 2 only allows to identify steady-state levels of the metabolites and no information about fluxes can be extrapolated from these values. Please provide pulse chase experiment with isotopic labeling if discussion of fluxes wants to be included in the manuscript.   

No statistics are provided for any of the changes on metabolite concentrations described in the manuscript.

How can the authors explain the differences between purine and pyrimidine changes? Might purine steady-state levels remain low but fluxes of their metabolic pathways increase?

The first 3 paragraphs of the discussion are redundant with the introduction.

The hypothesis that the cellular premise ´guilty-by-association´ stands in chlorovirus as put forward by the authors needs to be proven or removed from the manuscript. The authors provided no experimental evidence in their original publication 10.3390/v10100576.

Author Response

Dear reviewers,

A revised version of manuscript entitled " Early-phase drive to the precursor pool, chloroviruses dive into the deep-end of nucleotide metabolism" is attached.

We want to thank the reviewers for their thoughtful comments, which have improved the manuscript.  Our response to the reviewer's comments is attached and we hope the manuscript is now suitable for publication in Viruses. We hope the reviewers find that our corrections help clarify reviewers’ concerns.

Best personal regards,

David D. Dunigan

Reviewer 2 Report

The manuscript by Dunigan et al addresses the question of how the AT-rich chloroviruses can cope with the massive amount of nucleotides required for their replication in their GC-rich cellular host. The authors hypothesize that the virally-encoded genes involved in DNA synthesis reprogram the host cell metabolism for this purpose. They first thoroughly annotate the purine and pyrimidine metabolic pathway from host and cellular encoded enzymes. They also exploit their previously published transcriptomic data to analyze the expression timing of these enzymes along the virus replication cycle. Finally, they performed a metabolomic analysis of the pyrimidine metabolic pathway intermediates. Their results show that virally encoded enzymes can boost pyrimidine biosynthesis by opening new “routes” in the pyrimidine pathway using enzymes not encoded in the host’s genome. They also found viral genes complementing cellular encoded homologs, as well as viral genes that regulate the enzymatic cascade. 

Overall, I think this is a very interesting manuscript. The methods, analysis and interpretation are sound in my opinion. I would recommend this manuscript to be published. I think however that figures could be slightly reworked. 

For instance, I found Figures 1, 2 and 3 to be redundant. They could all be merged into a single figure or two. The lines, circles and arrows used to annotate Fig 2 and 3 are very bold and hide some neighboring enzymes names. 

In Table 2, the color used to express the fold change is not clear. The authors should add a color key with minimal and maximal values. 

In Fig S1 (A, B and C): what is the meaning of the polynomial regression? I think this is not used anywhere in the manuscript so I would simply remove it. In addition, some regression coefficients are pretty bad (see for instance ATP, dGTP and UMP).

In Fig 4. The confidence intervals should be put on the figure with error bars, instead of mentioning them in the legend.

Author Response

(The authors gave the same response as above.)

Author Response

(The authors gave the same response as above.)
